# Adult-Onset Neuronal Ceroid Lipofuscinosis in a Shikoku Inu

**DOI:** 10.3390/vetsci8100227

**Published:** 2021-10-12

**Authors:** Shinji Tamura, Masaya Tsuboi, Naotami Ueoka, Shoko Doi, Yumiko Tamura, Kazuyuki Uchida, Akira Yabuki, Osamu Yamato

**Affiliations:** 1Tamura Animal Clinic, 7-16 Yoshimien, Saeki-ku, Hiroshima 731-5132, Japan; d_s.mic.56@elf-petclinic.com (S.D.); mojo@lemon.megaegg.ne.jp (Y.T.); 2Laboratory of Veterinary Pathology, Graduate School of Agriculture and Life Science, University of Tokyo, 1-1-1 Yayoi, Bunkyo-ku, Tokyo 113-8657, Japan; atsuboi@g.ecc.u-tokyo.ac.jp (M.T.); auchidak@g.ecc.u-tokyo.ac.jp (K.U.); 3Ueoka Animal Hospital, 2 Chome-18-11 Yoshijimahigashi, Naka-ku, Hiroshima 730-0822, Japan; ueoka@ms6.megaegg.ne.jp; 4Laboratory of Clinical Pathology, Joint Faculty of Veterinary Medicine, Kagoshima University, 1-21-24 Korimoto, Kagoshima 890-0065, Japan; yabu@vet.kagoshima-u.ac.jp

**Keywords:** neuronal ceroid lipofuscinosis, Shikoku Inu, dog, MRI, retina

## Abstract

A two-year-and-eleven-month-old male Shikoku Inu was referred for evaluation of progressive gait abnormality that had begun three months prior. Neurological examination revealed ventral flexion of the neck, a wide-based stance in the hindlimb, wide excursions of the head from side to side, tremor in all four limbs, hypermetria in all four limbs, proprioceptive deficits in all four limbs, reduced patellar reflex in both hindlimbs, and postural vertical nystagmus. Later, behavioral and cognitive dysfunction, ataxia, and visual deficits slowly progressed. Magnetic resonance imaging revealed symmetrical progressive atrophy of the whole brain and cervical spinal cord. Bilateral retinal degeneration was observed, and both flush and flicker electroretinograms were bilaterally non-recordable at the age of five years and eight months, and the dog was euthanized. Histopathologically, faint-to-moderate deposition of light-brown pigments was frequently observed in the cytoplasm of neurons throughout the cerebrum, cerebellum, and nuclei of the brainstem. The pigments were positive for Luxol fast blue, periodic acid–Schiff, and Sudan black B, and exhibited autofluorescence. Electron microscopic examination revealed the accumulation of membranous material deposition in the neuronal cytoplasm. Small foci of pigment-containing macrophages were frequently observed around the capillary vessels. Based on these clinical and pathological findings, the animal was diagnosed with adult-onset neuronal ceroid lipofuscinosis.

## 1. Introduction

Neuronal ceroid lipofuscinosis (NCL) is a rare group of lethal neurodegenerative diseases characterized histopathologically by the abnormal accumulation of ceroid- or lipofuscin-like lipopigments in neurons and other cells throughout the body. These diseases are heritable as autosomal recessive traits. The major clinical signs are progressive motor, intellectual and behavioral dysfunction, epileptic seizures, and blindness [1].

NCL has been described in several canine breeds including English Setters [2], Dachshunds [3,4], Chinese Crested Dogs [5], Salukis [6], Labrador Retrievers [7], Border Collies [8,9,10], Cocker Spaniels [11], American Bulldogs [12], Australian Shepherds [13,14], Chihuahuas [15,16], Australian Cattle Dogs [17,18], Tibetan Terriers [19,20,21,22], Polish Owczarek Nizinny Dogs [22], American Staffordshire Terriers [23], Cane Corso Dogs [24], Alpenländische Dachsbracke Dogs [25], German Shorthaired Pointers [26], Golden Retrievers [27], Welsh Corgis [28], Yugoslavian Shepherds [29], Dalmatians [30], and a mixed breed [31]. The known causative mutations (*CLN1*/*PPT1* [4,24], *CLN2*/*TPP1* [3,22], *CLN5* [8,9,10,17,27], *CLN6* [13,14], *CLN7*/*MFSD8* [5,16], *CLN8* [25,26], *CLN10/CTSD* [12], *CLN12*/*ATP13A2* [18,19,20,21], *ARSG* [23]) have been identified in some of these. These identified genetic mutations each reside in different genes, and the clinical course can vary depending on the genetic mutation. This suggests that NCL in dogs is also variable and attributable to a number of different genes, as in humans, and is thought to be a model of human NCL [32].

In the present study, the clinical and pathological characteristics of adult-onset NCL observed in a Shikoku Inu, a traditional Japanese breed, are described. This is the first report of NCL in this breed.

## 2. Case Presentation

A two-year-and-eleven-month-old male Shikoku Inu, weighing 17.7 kg, was referred to the Tamura Animal Clinic for evaluation of progressive gait abnormality that had begun three months prior. Neurological examination revealed ventral flexion of the neck, a wide-based stance in the hindlimb, wide excursions of the head from side to side, tremor in all four limbs, hypermetria in all four limbs, proprioceptive deficits in all four limbs, reduced patellar reflex in both hindlimbs, and postural vertical nystagmus were observed. Routine laboratory testing revealed complete blood count and serum chemistry profiles with normal reference intervals, and right lateral and ventrodorsal thoracic radiography revealed no abnormalities. A blood smear stained with Wright–Giemsa stain demonstrated no abnormal cytoplasmic vacuoles in any type of leukocytes, including lymphocytes. Magnetic resonance imaging (MRI: 0.3T, AIRIS II comfort, FUJIFILM Healthcare, Tokyo, Japan) demonstrated symmetrical, diffuse, mild widening of the cerebral sulci, dilated fissures of the diencephalon, midbrain, and cerebellum, and mild lateral ventricular enlargement (Figure 1). Cerebrospinal fluid (CSF) was obtained via atlanto-occipital puncture, and revealed normal total nucleated cell count (<5/μL; reference level <5/μL) and a normal protein level (18.5 mg/dL; reference level <40 mg/dL). The serum immune-precipitation antibody titer against canine distemper virus (CDV) increased 6400-fold, the titer in the CSF increased less than 100-fold, and RNA of CDV in the CSF was negative. At that time, a degenerative disease was suspected. Activities of lysosomal β-hexosaminidase A (1199; reference level 1454 ± 401) and B (116; reference level 143 ± 85) were normal, suggesting that GM2 gangliosidosis had declined. No treatment was given according to the owner’s will. The clinical signs progressed gradually. At the age of three years and seven months, blindness, head pressing, unwillingness to walk, falling, absence of olfaction, mild decreased mental status, absence of bilateral menace response, and decreased nostril sensation were observed. Somnolence and difficulty walking were observed at four years old, astasia was observed at four years and seven months old, recumbent was observed at five years and three months old, and urinary incontinence was observed at five years and five months old. Follow-up MRIs were performed at the age of two years and eleven months, four years and seven months, and five years and eight months. MRI revealed progressive symmetrical atrophy of the whole brain and cervical spinal cord over time. The corpus callosum, especially in the caudal part, and the fornix became progressively thinner (Figure 1). Examination of the optic fundus performed at the age of five years and eight months revealed bilateral retinal degeneration, and both the flush and flicker electroretinograms were bilaterally non-recordable. The dog was euthanized at the owner’s request soon after the 3rd MRI examination because of crying at night, and necropsy was performed. The dog’s appetite remained normal until he was euthanized.

Histopathological examination was limited to the brain, eyeballs, and liver at the owner’s request. Symmetrical atrophy of the whole brain and cervical spinal cord, and the thinning of the corpus callosum and the fornix were observed macroscopically. Histopathologically, the ventricles of the brain were mildly dilated. Faint to moderate deposition of light-brown pigments was frequently observed in the cytoplasm of neurons throughout the cerebrum, cerebellum, and nuclei of the brainstem (Figure 2A–C). The accumulated pigments were positive for Luxol fast blue (LFB; Figure 2D), periodic acid–Schiff (PAS; Figure 2E), and Sudan Black B staining (Figure 2F). Autofluorescence was observed in unstained specimens under a fluorescence microscope, which was consistent with the neuronal pigments of the Purkinje cells (Figure 2G). These characteristics of the pigments were consistent with those of ceroid-lipofuscin. In addition to pigment deposition, slight neuronal loss was observed throughout the brain, especially in the molecular and Purkinje layers of the cerebellum (Figure 2B). Diffuse small vacuolation was observed in the neuropils of the cerebral and cerebellar white matter, and small foci of macrophages containing the pigments were frequently observed around the capillary vessels (Figure 2B,C). Meningeal thickening was not observed. In the orbits, the retina showed mild atrophy. The pigment was lightly deposited in the cytoplasm of the neuronal layer, and many macrophages were found at the border with the sclera, phagocytosing the pigment (Figure 2H). The liver was severely congested, and hepatocytes were mildly swollen with vacuolated cytoplasm, although lipofuscin deposition was not evident in the liver. Transmission electron microscopy of the formalin-fixed tissue revealed that the storage bodies in neurons from the cerebellar Purkinje cells consist of amorphous and membrane-like components (Figure 2I,J). Considering these findings and the age of onset, a diagnosis of adult-onset NCL was made.

Genomic DNA was extracted from whole blood of the dog. PCR and Sanger sequencing were conducted on two targeted areas, which were the locations of two mutations previously identified in canine *CLN12/ATP13A2* gene as the adult-onset NCL cause [18,19,20,21]. The sequence was shown to be homozygous wild-type and did not include those mutations.

## 3. Discussion

Among the many reports of NCL in dogs, adult-onset NCL was reported in Tibetan Terriers and Australian Cattle Dogs, both of which were reported to have causative mutations of the canine *CLN12/ATP13A2* gene: c.1623delG in Tibetan Terriers and c.1118C>G in Australian Cattle Dogs, respectively [18,19,20,21]. The neurological signs observed in Tibetan Terriers are relatively well documented. The first clinical signs were those of nyctalopia (night blindness) noted from two to three years of age. From four to six years of age, there were changes in behavior, characterized by loss of house training, confusion, fearfulness, occasional unpredictable aggressiveness, and poor-to-ravenous appetite. Motor signs of ataxia occurred late in the course of the disease and were severe in a 10-year-old affected dog, which was also dull and poorly responsive to visual, auditory, or tactile stimuli [19]. Tibetan Terriers with NCL often turn their heads from side to side repeatedly while standing still [19]. Neurological signs in Australian Cattle Dogs include anxiety, impaired ability to recognize and respond to previously learned commands, increased sensitivity to loud or unexpected sounds, sleep disturbances, inappropriate persistent vocalization, impaired ability to navigate stairs and to jump up or down from furniture, trembling, seizures, stiffness or weakness, loss of coordination, and ability to be seen in both bright and dim light [18].

In addition, ataxia, proprioceptive deficit, pacing, reduced palpebral reflex and menace response, normal pupillary light reflexes in response to bright stimuli, and aggressiveness were observed, although spinal reflexes were normal. These symptoms occurred at approximately 6 years of age, and progressively worsened over time; the dogs were euthanized at 7 to 9 years of age [18]. The neurological signs observed in the present Shikoku Inu with NCL were similar to those in these two breeds in terms of adult onset of slowly progressive behavioral and cognitive dysfunction, ataxia, and visual deficits. However, they differ in that ventral flexion of the neck, postural vertical nystagmus, reduced patellar reflex, and absence of aggressiveness were observed in the present Shikoku Inu.

The MRI feature of the present Shikoku Inu is progressive atrophy of the whole brain and cervical spinal cord, without abnormal signal intensity. MRI of cases of canine NCL have been reported in Border Collies [8,10], Dachshunds [3], Chihuahuas [15], and Tibetan Terriers [20]. Atrophy of the cerebrum, diencephalon, and cerebellum was a common finding. In addition, subdural hematoma was reported in a Dachshund [3], and meningeal thickening and enhancement were reported in Chihuahuas [15]. In humans, diffuse atrophy of the whole brain, thinning of the cerebral cortex, and deep white matter T2W-hyperintensities reflect demyelination and gliosis (leukoencephalopathy), and T2W-hypointensities in the thalamus and basal ganglion of unknown mechanisms have been reported [33,34,35]. There have been only a few cases of canine NCL examined with MRI; therefore, it is unknown why no abnormal signal intensities, as observed in humans on MRI, were reported in canine NCL. Abnormal findings, such as thinning of the corpus callosum on midline sagittal images, have been reported in lysosomal storage diseases, including late juvenile or early adult-onset NCL in Border Collies and Chihuahuas [36]. Progressive thinning of the corpus callosum and fornix was also observed on MRI in the present Shikoku Inu.

The neuropathological features observed in the present case are consistent with canine NCLs in other breeds, with moderate lipofuscin deposition in the neurons and pigment-phagocytosing macrophage infiltration in the perivascular region. The immunohistochemical and ultrastructural features of the deposits were similar to those of previous adult-onset NCL reports in Australian Cattle Dogs [18] and Tibetan Terriers [20]. When compared to NCL in other breeds, the severity of the brain lesions was relatively mild, neuronal loss in the brain was sparse, and gliosis was not evident in the present case. The difference in severity may be attributed to the slow progression of the disease in the present case. Interestingly, the degree of neuronal loss and lipofuscin deposition in the present case was minimal, even when compared to adult-onset NCLs in Tibetan Terriers and Australian Cattle Dogs. Since this case was a post-euthanasia necropsy case, it is possible that we were observing an earlier histopathology. The reasons hypothesized for these minor lesions need to be verified by accumulating more cases. In addition to the brain, lipofuscin deposition was also observed in the retina, but not in the hepatocytes or circulating white blood cells. It is hypothesized that cell degeneration might be confined to neuronal tissues, but this hypothesis needs to be verified by investigating the rest of the organs.

The dog did not have the two mutations in the canine *CLN12/ATP13A2* gene that have been reported in Tibetan Terriers and Australian Cattle Dogs with adult-onset NCL, suggesting that this Shikoku Inu is a new adult-onset NCL. The limitation is that the clinical and pathological considerations are based on only the one case. Further studies are needed to clarify this new type of canine NCL using additional affected cases in Shikoku Inus. As a next step, we wish to identify the causative gene of NCL in Shikoku Inus.

## 4. Conclusions

In the present study, we describe the clinical and pathological characteristics of adult-onset NCL observed in Shikoku Inu. This is the first report of NCL in this breed. General clinical signs, MRI findings, and pathological findings observed in the present Shikoku Inu with NCL were similar to those in other breeds with adult-onset NCL. However, they differed in some respects, including the severity of the brain lesion being relatively mild, which may be attributed to the slow progression.

## Figures and Tables

**Figure 1 vetsci-08-00227-f001:**
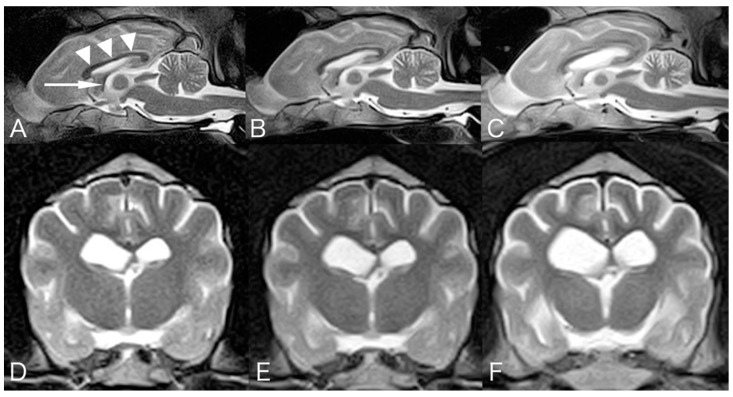
Mid-sagittal T2-weighted magnetic resonance images at the age of 2 years and 11 months (**A**), 4 years and 7 months (**B**), and 5 years and 8 months (**C**). Transverse T2-weighted MR images at the level of the interventricular foramen at the time of 2 years and 11 months (**D**), 4 years and 7 months (**E**), and 5 years and 8 months (**F**). Progressive symmetrical atrophy was observed in the forebrain, cerebellum, brainstem, and cranial cervical spinal cord. Corpus callosum (arrowhead) and fornix (arrow) show gradual thinning.

**Figure 2 vetsci-08-00227-f002:**
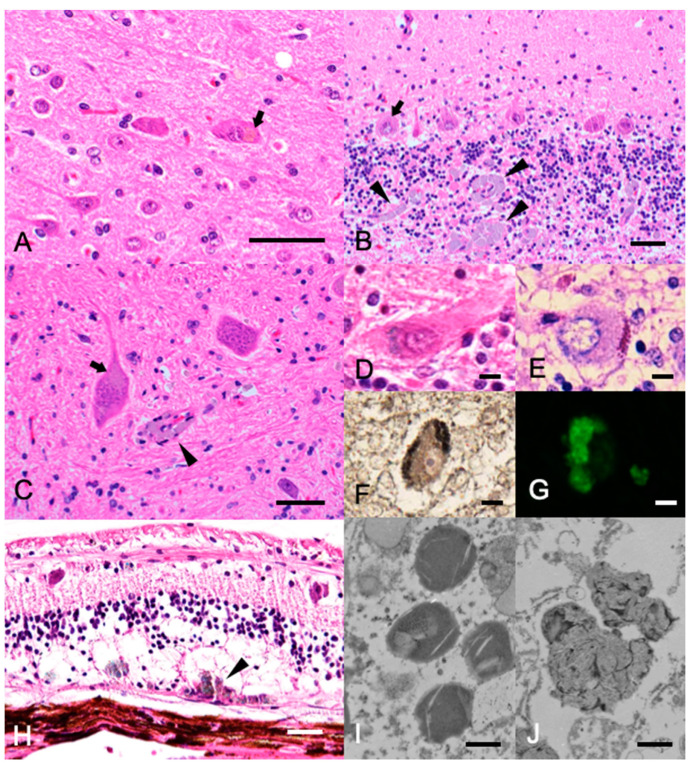
Histopathology of the (**A**–**C**) Hematoxylin and eosin stain of the cerebrum (**A**), cerebellum (**B**), and neuronal nuclei of the pons (**C**). Bars = 50 µm. Light-brown pigments were frequently observed in the cytoplasm of large neurons (arrows). Note the pigment-phagocytosing macrophage infiltration in the perivascular region (arrowheads). (**D**–**F**) Histochemical staining of the cerebellum. Bars = 10 µm. The accumulated pigments in the Purkinje cells appeared blue in Luxol fast blue-hematoxylin and eosin (LFB-HE) stain (**D**), purple in periodic acid–Schiff (PAS) stain (**E**), and black in Sudan Black B stain (**F**). (**G**) Fluorescence microscopy image of the cerebellum. The accumulated pigments in the Purkinje cells showed green autofluorescence. (**H**) LFB-HE stain of the retina. Bar = 50 µm. The pigments were observed in the cytoplasm of the neuronal layer, and many macrophages phagocytosing the pigment were found at the border with the sclera (arrowheads). (**I**,**J**) Transmission electron microscopy of the storage bodies in neurons from the cerebellar Purkinje cells. Bar = 0.5 µm. The bodies showed membrane-like components.

## Data Availability

Data sharing not applicable.

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
