# Peer review of "Adult-Onset Neuronal Ceroid Lipofuscinosis in a Shikoku Inu"

_vetsci, 2021, doi:10.3390/vetsci8100227_

Round 1

Reviewer 1 Report

Dear authors,

Thank you for submitting an interesting case report of neuronal ceroid lipofuscinosis in a new breed. The MRI changes over the progression of the disease is certainly very fascinating. There were several places where the subject-verb agreement was incorrect and a few sentences that I had trouble following. I think detailed copy editing would be helpful.

Therefore, several points should be first addressed before considering it for publication.

Abstract

The neurological examination needs to be amended as part of the description does not follow the proper terminology:

  • Head tilt to both sides
  • Ventral flexion tail
  • Amble
  • Depression of the patellar reflex in both hindlimbs

I would suggest deleting amble and ventral flexion of the tail and use wide excursions of the head from side to side and reduced or absent patellar reflex instead.

Introduction

Line 41 - psychointellectual is human terminology please use appropriate terminology. And also amend seizures for epileptic seizures.

Line 42 – several most recently reported breeds are missed in the manuscript (e.g German Shorthaired Pointer, Golden retriever, Welsh Corgi, Yugoslavian Shepherd, Dalmatian and/or a mixed breed) Please add the appropriate references.

Lines 47-49 – Please update the references for the current list of known causative mutations.

Lines 51-52 – The current classification for canine NCL includes a juvenile and adult-onset forms of the disease. Please amend and delete the current reference as based in an old human classification.

Case presentation

Please re-write the neurological examination following the recommendations in the abstract section.

Line 67 – please state the views used in the thoracic radiography.

Line 69 – Please mention the MRI characteristics (e.g 1.5 T) and the brand of the machine.

Lines 69-71 – Please amend the MRI findings – widening instead of dilation of the cerebral sulci; diencephalon instead of diencephalons; and add mild in lateral ventricular enlargement.

Line 72 – Please amend cell numbers for total nucleated cell count

Line 78 – The sentence is difficult to follow. Please amend.

Lines 89-90 – The sentence is difficult to follow. Please amend.

Figure 1

Line 100 – cranial instead of rostral. The arrowhead and the arrows are missed in the figure.

Discussion

I think detailed copy editing of parts of the discussion would be helpful:

  • Lines 138-141
  • Please re-write the neurological examination following the recommendations in the abstract section.
  • Lines 174-76
  • Lines 191-93

Line 198 – the lack of mutation found is not a limitation of the study

References

Please amend the references based on the above recommendations.

Author Response

First of all, we appreciate the time the editor and reviewers have taken to read and review our manuscript. Their valuable comments have significantly improved and clarified several aspects of our paper. The following document presents our responses to comments from the reviewers. It includes the original comments in italics and the subsequent responses we made.  

Abstract

The neurological examination needs to be amended as part of the description does not follow the proper terminology:

Head tilt to both sides

Ventral flexion tail

Amble

Depression of the patellar reflex in both hindlimbs

I would suggest deleting amble and ventral flexion of the tail and use wide excursions of the head from side to side and reduced or absent patellar reflex instead.

RESPONSE: The authors deleted amble and ventral flexion of the tail and used wide excursions of the head from side to side and reduced patellar reflex instead.

Introduction

Line 41 - psychointellectual is human terminology please use appropriate terminology. And also amend seizures for epileptic seizures.

RESPONSE: The authors changed to ‘intellectual and behavioral dysfunction’ and added ‘epileptic’.

Line 42 – several most recently reported breeds are missed in the manuscript (e.g German Shorthaired Pointer, Golden retriever, Welsh Corgi, Yugoslavian Shepherd, Dalmatian and/or a mixed breed) Please add the appropriate references.

RESPONSE: The authors added reported those breeds.

Lines 47-49 – Please update the references for the current list of known causative mutations.

RESPONSE: The authors updated the references.

Lines 51-52 – The current classification for canine NCL includes a juvenile and adult-onset forms of the disease. Please amend and delete the current reference as based in an old human classification.

RESPONSE: The authors delete the sentences and changed reference No.1.

Case presentation

Please re-write the neurological examination following the recommendations in the abstract section.

RESPONSE: The authors re-wrote.

Line 67 – please state the views used in the thoracic radiography.

RESPONSE: The authors added ‘right lateral and ventrodorsal’.

Line 69 – Please mention the MRI characteristics (e.g 1.5 T) and the brand of the machine.

RESPONSE: The authors added ‘0.3T, AIRIS â…¡ comfort, FUJIFILM Healthcare, Japan’.

Lines 69-71 – Please amend the MRI findings – widening instead of dilation of the cerebral sulci; diencephalon instead of diencephalons; and add mild in lateral ventricular enlargement.

RESPONSE: The authors implemented the recommended changes.

Line 72 – Please amend cell numbers for total nucleated cell count

RESPONSE: The authors changed to ‘total nucleated cell count’.

Line 78 – The sentence is difficult to follow. Please amend.

RESPONSE: The authors changed to “No treatment was given according to the owner's will. The clinical signs progressed gradually.”

Lines 89-90 – The sentence is difficult to follow. Please amend.

RESPONSE: The authors changed to “Follow-up MRIs were performed at the age of two-years-and-eleven-months, four-years-and-seven-months, and five-years-and-eight-months. MRI revealed progressive symmetrical atrophy of the whole brain and cervical spinal cord over time. The corpus callosum, especially in the caudal part, and the fornix became progressively thinner.”

Figure 1

Line 100 – cranial instead of rostral. The arrowhead and the arrows are missed in the figure.

RESPONSE: The authors changed to ‘cranial’ and added the arrowhead and the arrow.

Discussion

I think detailed copy editing of parts of the discussion would be helpful:

We submitted our original paper to an English editing company (Editage), but we can understand that there are still some parts that need to be revised as the reviewer mentioned. So, we revised a couple of sentences. If there are still the parts that should be revised, please point out where and what they should be. We are ready to submit our revised paper for English editing again.

Lines 138-141

Please re-write the neurological examination following the recommendations in the abstract section.

RESPONSE: The authors re-wrote.

Lines 174-76

Lines 191-93

Line 198 – the lack of mutation found is not a limitation of the study

RESPONSE: The authors removed the sentence.

References

Please amend the references based on the above recommendations.

RESPONSE: The authors amend the references.

Reviewer 2 Report

In the current review manuscript “vet sci-1388892”, the authors conduct an interesting case study examining the neuronal ceroid lipofuscinosis in the Shikoku Inu breed. The authors report clinical exam findings such as young-onset, progressive neurological deficits, behavioral, cognitive aspects, MRI findings revealing atrophy in the cerebrum and spinal cord. Histopathology evidence is also presented. Overall, the study is well conducted and the topic is generally of interest to the readers of the journal. I have made a few suggestions that I believe will improve the overall quality of the manuscript.

The manuscript does provide a good layout (short introduction, case presentation, and discussion) yet some of the MRI findings are unclear and if the lesions were symmetrical vs asymmetrical and corresponding pathological observations.

The emerging literature in this field especially regarding additional breeds could be included.

The authors have a discussion focussed on CLN12/ATP13A2 gene mutations. However, screening for this mutation or any other relevant mutations pertaining to this study is highly desirable.

A supplementary methods section if allowed for the case report is advised to reproduce the details of the reagents, protocols.

I recommend discussing the limitations associated with this model and potential clinical considerations.

Author Response

First of all, we appreciate the time the editor and reviewers have taken to read and review our manuscript. Their valuable comments have significantly improved and clarified several aspects of our paper. The following document presents our responses to comments from the reviewers. It includes the original comments in italics and the subsequent responses we made.

The manuscript does provide a good layout (short introduction, case presentation, and discussion) yet some of the MRI findings are unclear and if the lesions were symmetrical vs asymmetrical and corresponding pathological observations.

RESPONSE: The authors added ‘Symmetrical’ at the MRI findings part, and ‘Symmetrical atrophy of the whole brain and cervical spinal cord, and the thinning of the corpus callosum and the fornix were observed macroscopically ’at the pathological part.

The emerging literature in this field especially regarding additional breeds could be included.

RESPONSE: The authors added literatures about German Shorthaired Pointer, Golden retriever, Welsh Corgi, Yugoslavian Shepherd, Dalmatian and a mixed breed.

The authors have a discussion focussed on CLN12/ATP13A2 gene mutations. However, screening for this mutation or any other relevant mutations pertaining to this study is highly desirable.

RESPONSE: The authors added the description related to the preliminary molecular analysis and result in the Case Presentation section “Genomic DNA was extracted from whole blood of the dog. PCR and Sanger sequencing were conducted on two targeted areas, which were the locations of two mutations previously identified in canine CLN12/ATP13A2 gene as the adult-onset NCL cause [18-21]. The sequence was shown to be homozygous wild-type and did not include those mutations.” and in the Discussion section “The dog did not have the two mutations in canine CLN12/ATP13A2 gene that have been reported in Tibetan terriers and Australian cattle dogs with adult-onset NCL, suggesting that this Shikoku Inu is a new adult-onset NCL.”

A supplementary methods section if allowed for the case report is advised to reproduce the details of the reagents, protocols.

RESPONSE: The authors added Supplementary Materials data regarding MRI.

I recommend discussing the limitations associated with this model and potential clinical considerations.

RESPONSE: The authors added “The limitation is that the clinical and pathological considerations are based on the only one case. Further studies are needed to clarify this new type of canine NCL using additional affected cases in Shikoku Inus.”
